# Effects of Shrub Encroachment in the Inner Mongolia Ecotones between Forest and Grassland on the Species Diversity and Interspecific Associations

**Qing Song and Tiemei Wang \***



School of Grassland Science, Beijing Forestry University, Beijing 100083, China
* Correspondence: alfalfa@126.com; Tel.: +86-10-6233-8527

**Abstract:** Shrub encroachment, which is the increase in shrubs or woody plants in grassland, is one of the important ecological problems facing grassland worldwide. The study was conducted in the ecotones between the forest and grassland of Inner Mongolia. Using a simple random sampling method, 40 shrubland sample plots, evenly distributed in the typical forest-grass transition area, were selected for community investigation. According to the steppe shrub encroachment index, the grassland was divided into different levels. The species diversity and interspecific association of different levels of sample plots were statistically analyzed and compared. It was found that the Shannon-Wiener index (H) and Simpson index (P) were negatively associated with the shrub encroachment index ($p < 0.01$) and gradually decreased with increasing levels of shrub encroachment. When the grassland transitioned to severe shrub formation, the species diversity of the community significantly decreased, and H and P were significantly lower than those of the mild and moderate shrub-steppe, and the lowest values were 1.37 and 0.48, respectively. With the increase in the shrub encroachment index, biomass showed a trend of first increasing and then decreasing. The aboveground biomass of shrub interspace and total aboveground biomass in the severe shrub encroachment steppe were significantly lower than those in the mildly and moderately shrub-steppe, with the lowest values of 101.86 g/m$^2$ and 189.69 g/m$^2$, respectively. Shrub encroachment led to a change in the overall association of shrub communities in the ecotones between the forest and grassland of Inner Mongolia from positive to negative. The vast majority of species pairs in the shrub community showed no significant association, and the interspecies association was relatively weak. The results showed that shrub encroachment would affect community species diversity; mild and moderate shrub encroachment had no significant impact on community species diversity, but severe shrub encroachment significantly reduced the community species diversity. There was no significant effect of shrub encroachment on aboveground biomass, which increased first and then decreased, and the herbaceous plant population played a leading role in grassland productivity. The interspecific association of grassland is loose and weak. The community was in the early stage of unstable succession, and it was possible to continue shrub encroachment or reverse succession into a typical grassland in response to the interference of human or environmental factors.

**Keywords:** shrub encroachment; ecotones between forest and grassland; species diversity; interspecific association



## 1. Introduction

Grassland is an important part of terrestrial ecosystems and is also characterized by harsh natural environments and poor ecological stability, which affects open, shrubby, and savanna grasslands. Grassland accounts for approximately 40% of the global surface area and 69% of the agricultural land area [1]. In arid and semiarid grassland areas, climate change and inappropriate land utilization have led to an increase in the density, coverage, and biomass of shrubs, a phenomenon known as shrub encroachment [2,3]. Shrub encroachment in grasslands is common in various ecosystem types in Africa, the Americas,

Australia, Asia, and coastal countries and regions of the Mediterranean, including humid grasslands, polar tundra, and arid and semiarid grasslands [4]. Approximately 10 to 20% of the world's arid and semiarid grasslands are undergoing shrub encroachment processes [5]. Shrub encroachment in China is mainly found in the warm grassland of the Inner Mongolia Plateau and Loess Plateau and the alpine grassland in the northeast of the Qinghai-Tibet Plateau [6]. Shrubs have a significant effect on the species composition, biodiversity, and grassland productivity of the plant community, interfere with the structure and stability of the grassland plant community, change the spatial pattern of the original vegetation, and reduce species richness. Their encroachment threatens the function and service supply of grassland ecosystems [7], affects soil water infiltration [8], and accelerates soil drought [9]. Severe cases may lead to land degradation and desertification in arid and semiarid areas [10].

In arid and semiarid areas, shrubs can directly or indirectly affect the herb community under the canopy by regulating soil moisture and nutrients, improving the temperature and radiation required by plant growth in the canopy, and creating a favorable microenvironment [11,12]. On the scale of a sample plot, due to the interference of climatic factors and human factors, the effect of shrubs on the biodiversity of grassland ecosystems is uncertain. In the grassland ecosystem of most semiarid areas, when shrub vegetation occupies a dominant position, the herb community gradually evolves into a shrub-herb community to adapt to the new environment, and the area of the grassland that is covered by shrubs increases [13,14]. When the dominant species change from herbaceous plants to woody plants, the species diversity of the community also changes. Ding Wei et al. [15] found that annual and biennial herbaceous plants significantly increased in the shrub community, which enriched the diversity of herbaceous species to a certain extent. Wang Yingxin et al. [16] believed that on the patch scale, the fertile island effect was conducive to the accumulation of nutrients and water under shrubs, providing a good environment for plant growth. Guo Pu et al. [17] found that increases in the percentage of shrubs affected the species evenness of herbaceous plants and had a negative impact on the diversity. Ratajczak et al. [18] analyzed 13 different grassland or savanna communities in North America and found that when the area covered by shrubs expanded, the species richness in the ecosystem decreased by 45% on average. Some studies have shown that the effect of shrubbery on grassland ecosystems was mixed [19]. Shrubbery provides richer litter and a warmer microclimate. Low shrubs prevent livestock from eating too much and protect understory herbaceous species, which positively affects them. However, in the process of shrub encroachment, the shade effect of undergrowth herbaceous species is increased, and the loss of occasional and weakly light-competitive species in steppe plant communities leads to a reduction in the richness. The species diversity of the community is also affected by livestock trampling on tall shrubs with weak grazing resistance [20,21]. Some studies have reported that on the patch scale, shrub patches are significantly dominant in growth space competition with herbaceous plants, reducing the richness of herbaceous species; however, on the sample plot scale, shrubs lead to the fragmentation of the sample plot landscape, causing niche differentiation, which may increase species richness [22].

Aboveground herbaceous plant biomass is an important indicator of the ecological function of grasslands, and shrubs are usually regarded as a sign of grassland ecosystem degradation [23]. Grassland shrub encroachment occupies the growth space of herbaceous plants, and the growth environment changes, limiting key resources such as light and water needed by herbaceous plants, and the increase in shrub coverage significantly reduces the aboveground biomass of herbaceous plants [24]. Studies have found that shrubs have a higher leaf area index, and shrubs absorb only the top 0.7% of the light at the bottom of the light spectrum, resulting in herbage productivity decline [25]. In recent years, an increasing number of studies have shown that a certain degree of shrub coverage is beneficial to the accumulation of soil nutrients and can increase the productivity of grassland ecosystems [26]. Compared with herbaceous plants, several shrub species have poor palatability, and prickly shrubs are not easily eaten by livestock; they have strong

grazing tolerance and protect the plants under them [27,28]. For these reasons, shrub patches maintain higher biomass than grassland patches, and this characteristic is not affected by grazing intensity [19]. Peng et al. [29] found that in scrubbed grasslands in Inner Mongolia, China, the vegetation cover of shrub patches increased. Zhao et al. [30] also showed that shrub encroachment significantly increased the aboveground vegetation biomass in desert grasslands. Thus, the effects of shrub encroachment on the community composition of the ecosystem, species diversity, and aboveground biomass are variable based on the region and study scale.

The grassland of the Inner Mongolia Plateau is located in the arid, semiarid, and semi-humid areas of central Asia and plays an important role in maintaining ecological stability and biodiversity in northern China. In recent years, due to natural and human factors, the grassland of the Inner Mongolia Plateau has undergone serious degradation and loss of biodiversity, and the encroachment of shrubs is considered to be a process. In some regions, climate change leads to a decrease in soil surface water content and an increase in $CO_2$ concentration in the atmosphere, which reduces the competition ability of herbaceous species and expands the invasion of shrubs [13,31]. Selective feeding of livestock caused by grazing will also give some shrubs a competitive advantage. In addition, the influence of climate or human activities leads to a decrease in grassland burning frequency, which puts shrubs in a favorable position in competition with herbaceous plants [7]. These factors cause shrub encroachment, which reduces the livestock-carrying capacity of grassland and affects production and ecological function. As the ecotone of the two vegetation types, the forest-grass transition zone is very sensitive to climate change and human disturbance, and it is also an important ecological barrier in the north.

The present study aimed to evaluate the effects of different levels of shrub encroachment on species diversity and interspecific associations. In this study, the shrub steppe vegetation in the forest–grass transition area of Xing'an, Inner Mongolia, was taken as the research object. The representative shrub steppe was selected for study, and the relationship between the shrub index, biodiversity index, and interspecific association was analyzed. This can provide a basis for the scientific evaluation of the health and stability of grassland ecosystems in the Inner Mongolia forest-grass transition zone and further reveal the effects of shrub encroachment on grassland ecosystem stability and ecological function.

## 2. Materials and Methods

### 2.1. Overview of the Research Area

In this study, Xing'an League was selected as the study area. It is located in the northeastern Inner Mongolia Autonomous Region. The geographical location is 119°28′ to 123°39′ E, 44°15′ to 47°39′ N, and the altitude is approximately 150 to 1800 m. Mountains and hills account for approximately 95% of the area, and plains account for approximately 5%. From northwest to southeast, the corresponding vegetation is forestland, grassland, and crops, representing a typical forest–grass transition zone and pastoral farming ecotone. The area belongs to the temperate continental monsoon climate zone, with an annual average temperature of 4 to 6 °C in most areas and −3.2 °C in the northwest forest region, with an average annual precipitation of 400 to 450 mm. In the grassland plant community, herbaceous plants mainly include *Leymus chinensis*, *Cleistogenes squarrosa*, *Carex appendiculata*, *Sanguisorba officinalis*, and *Potentilla chinensis*. Shrub species in grassland areas mainly include *Prunus sibirica*, *Ostryopsis davidiana*, and *Spiraea salicifolia*.

### 2.2. Survey of Sample Plots

Using the simple random sampling method, 40 shrubland sample plots evenly distributed in the typical forest-grass transition area were selected as investigation areas (Figure 1). The survey time was from July to August 2021. The topography, slope position, and soil texture of the sample plots were observed and recorded, and a 10 × 10 m shrub quadrat was set up in each plot to investigate the ecological characteristics of the community. The number of shrubs in the quadrat was recorded. The height and length

of the major and minor axes of each shrub in the quadrat were measured, and the crown width of the shrubs was calculated according to the elliptic area formula. Shrub coverage was the ratio of the crown area of all shrub plants in the quadrat to the quadrat area. The newborn branches of the shrubs grown during the year of the survey in the quadrat were cut, naturally air-dried to a constant weight, and then weighed to obtain the aboveground biomass data. Three 1 × 1 m herbaceous quadrats among shrub interspaces were randomly selected in each shrubland sample plot. The total coverage of the community was estimated. The species and the number of plant clusters in the quadrats were recorded, and the density of each species was calculated. The species in the herbaceous quadrats were separately cut, and the litter was collected. They were air-dried to a constant weight and weighed to obtain aboveground biomass data.

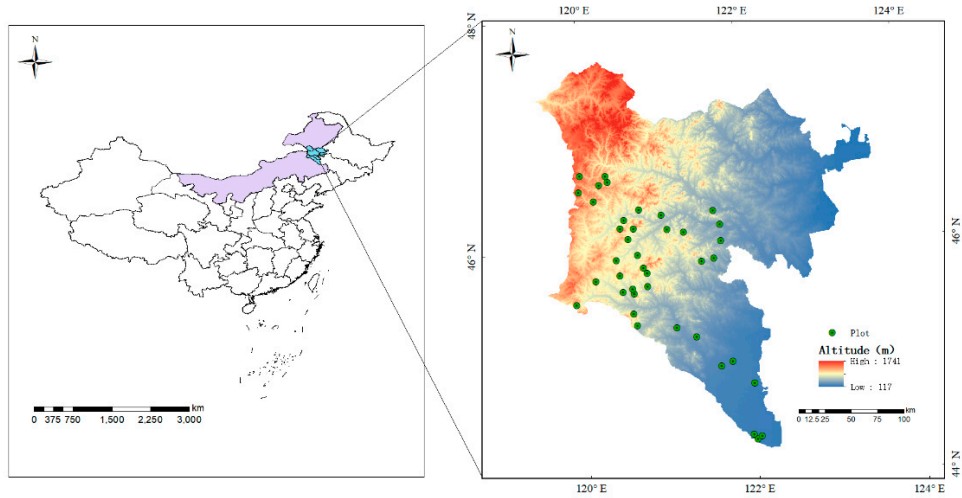

**Figure 1.** The distribution map of shrub plots in the forest-grass transition area of Inner Mongolia.

*2.3. Statistical Analysis of Data*

2.3.1. Calculation of Important Value and Species Diversity Index

The importance value (IV) reflected the dominance degree of species in the community. The IV of each species was calculated with the plot as the unit, and, according to the results of the field investigation, the dominant herbaceous species were selected with the IV of species as the evaluation index.

The IV is calculated as follows:

$$IV = (\text{relative density} + \text{relative coverage} + \text{relative height} + \text{relative frequency})/4 \quad (1)$$

The species diversity index of the herb community in shrubland included the Margalef (R), Shannon-Wiener (H), Simpson (P), and Pielou (E) indices. The calculation methods can be found in reference [32].

2.3.2. Calculation of the Steppe Shrub Encroachment Index

In this study, shrub and herb community characteristics were used as the main basis for classification [33,34] to calculate the steppe shrub encroachment index (SSEI). The formula is as follows:

$$SSEI = [(\text{relative density of shrubs} + \text{relative coverage of shrubs} + \text{relative biomass of shrubs})/3 \times 1 \times R] \times 100 \quad (2)$$

2.3.3. Overall Association Test

The variance ratio (*VR*) method proposed by Schluter [35] was used to calculate the overall association of species within the sample plots of different levels of shrub

encroachment, and the overall association was tested for significance by the statistic $W$, calculated as follows:

$$\delta^2 = \sum_{i=1}^{S} P_i(1 - P_i) \tag{3}$$

$$P_i = n_i/N \tag{4}$$

$$S_T^2 = \frac{1}{N} \sum_{i=1}^{N} (T_i - t)^2 \tag{5}$$

$$VR = S_T^2/\delta_T^2 \tag{6}$$

$$W = VR \times N \tag{7}$$

In the formula, $\delta^2$ is the total sample variance, that is, the variance of the frequency of all herbaceous species, $S^2$ is the variance of the number of species in all quadrats, $S$ is the total number of species, $N$ is the total number of quadrats, $P_i$ is the frequency of species $i$, $N_i$ is the number of quadrats of species $i$, $T_j$ is the number of species in quadrat $j$, and $t$ is the average number of species in all quadrats. $VR$ is the overall association index between the main species at all levels, and $VR = 1$ indicated that there was no association among all species; if $VR > 1$, there was a positive association among species, while if $VR < 1$, there was a negative association. $W$ was the statistic used to test the significance of $VR$. If $W > x^2_{0.05\ (N)}$ or $W < x^2_{0.95\ (N)}$, the overall association between species was significant; in contrast, if $x^2_{0.95\ (N)} < W < x^2_{0.05\ (N)}$, the overall association between species was not significant.

### 2.3.4. Interspecies Association Analysis

The $\chi^2$ statistic based on the 2 × 2 contingency table is used to qualitatively study the interspecific association. Before the $\chi^2$ test, the original data is converted into a binary data matrix according to whether the species exists in the plot, and then based on the binary data matrix, the qualitative data of all species pairs of the main population are listed in a 2 × 2 joint list, calculating the value of $a$, $b$, $c$ and $d$. The discontinuity of sampling leads to low estimation, which can be corrected using Yates' continuous correction coefficient, for which the formula is:

$$\chi^2 = \frac{N[|ad - bc| - 0.5N]^2}{(a + b)(a + c)(c + d)(b + d)} \tag{8}$$

In the formula, $a$ is the number of quadrats where both species appear, $b$ is the number of quadrats where species B is present but species A is absent, $c$ is the number of quadrats where species A is present, but species B is absent, and $d$ is the number of quadrats in which neither species is present. Generally, when $\chi^2 > 6.635$, that is, $p < 0.01$, the interspecific association is considered to be extremely significant; when $3.841 < \chi^2 < 6.635$, that is, $0.01 < p < 0.05$, the interspecific association is considered significant; when $\chi^2 < 3.841$, that is, $p > 0.05$, the interspecific association is considered to be insignificant. When $ad < bc$, it indicates a negative association, and when $ad > bc$, it indicates a positive association.

### 2.3.5. Interspecies Association Determination

The $\chi^2$ test of interspecific association only considered the presence of species in the quadrat without considering their abundance. It can be used to quickly determine the stability of the community and the prominent interspecific relationship, but the difference between the associations can be blurred. The Spearman rank association analysis is a non-parametric test and does not require the distribution form of species and can clarify the basic status of the spatial distribution of plants, based on a scientific and logical framework [36,37]. In this study, the importance value (IV) was introduced as a quantitative

indicator, and the Spearman association coefficient was used to measure the strength of interspecific associations. It is calculated as follows:

$$r(i,j) = 1 - \frac{6 \sum_{k=1}^{N} \left( x_{ik} - x_{jk} \right)^2}{N^3 - N} \tag{9}$$

In the formula, $r(i,j)$ is the Spearman rank association coefficient between species $i$ and $j$ in quadrat $k$, $r(i,j) \in (1, 1]$; a positive value indicates positive association, and a negative value indicates negative association; $N$ is the total number of plots; $x_{ik}$ and $x_{jk}$ are the ranks of species $i$ and $j$ in plot $k$ (calculated by IV), respectively.

## 3. Results

### 3.1. Effects of Shrubs on the Herb Community Species Diversity Index

The grasslands based on the steppe shrub encroachment index (SSEI) can be classified into three levels (Table 1, Figure 2): level I, mild shrub encroachment, $0 \leq$ SSEI < 5; level II, moderate shrub encroachment, $5 \leq$ SSEI < 15; level III, severe shrub encroachment, $15 \leq$ SSEI < 30.

**Table 1.** The classification of shrubland plots.

| Shrub Encroachment Level | SSEI | Plot Number | Shrub Coverage % | Shrub Abundance Plant/m² | Average Shrub Biomass g/m² | Herb Community Margalef Index |
|---|---|---|---|---|---|---|
| Level I | $0 \leq$ SSEI < 5 | 10 | 1.11–25.12 | 0.02–0.32 | 0.3–116.82 | 33.42–85.47 |
| Level II | $5 \leq$ SSEI < 15 | 18 | 1.16–54.12 | 0.07–0.55 | 1.25–743.92 | 10.69–76.06 |
| Level III | $15 \leq$ SSEI < 25 | 12 | 1.62–98.36 | 0.02–0.43 | 16.93–297.79 | 4.39–12.27 |

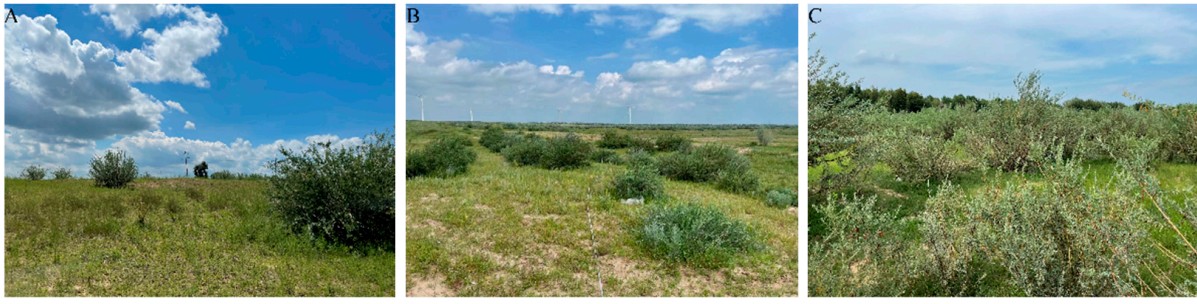

**Figure 2.** I-level (**A**), II-level (**B**), and III-level (**C**) shrub grasslands in the forest–grass transition zone of Inner Mongolia (*Caragana korshinskii* as an example).

The fitted curve representing the relationship between the steppe shrub encroachment index and the herb community diversity index and the differences in species diversity of the herb community in different levels of shrub encroachment in grassland is shown in Figure 3. The Shannon-Wiener index (H) and Simpson index (P) were significantly negatively associated with the SSEI and decreased with increasing shrub degree. The H value of the level I shrub steppe ($0 \leq$ SSEI < 5) was generally higher, and the maximum value was 2.64. The H value of the level III shrub steppe ($15 \leq$ SSEI < 25) was lower, and the minimum value was only 0.68. The $p$ value of the grade I shrub steppe was higher, and the maximum value was 0.92. The $p$ value of the level III shrub steppe was lower, and the minimum value was 0.48.

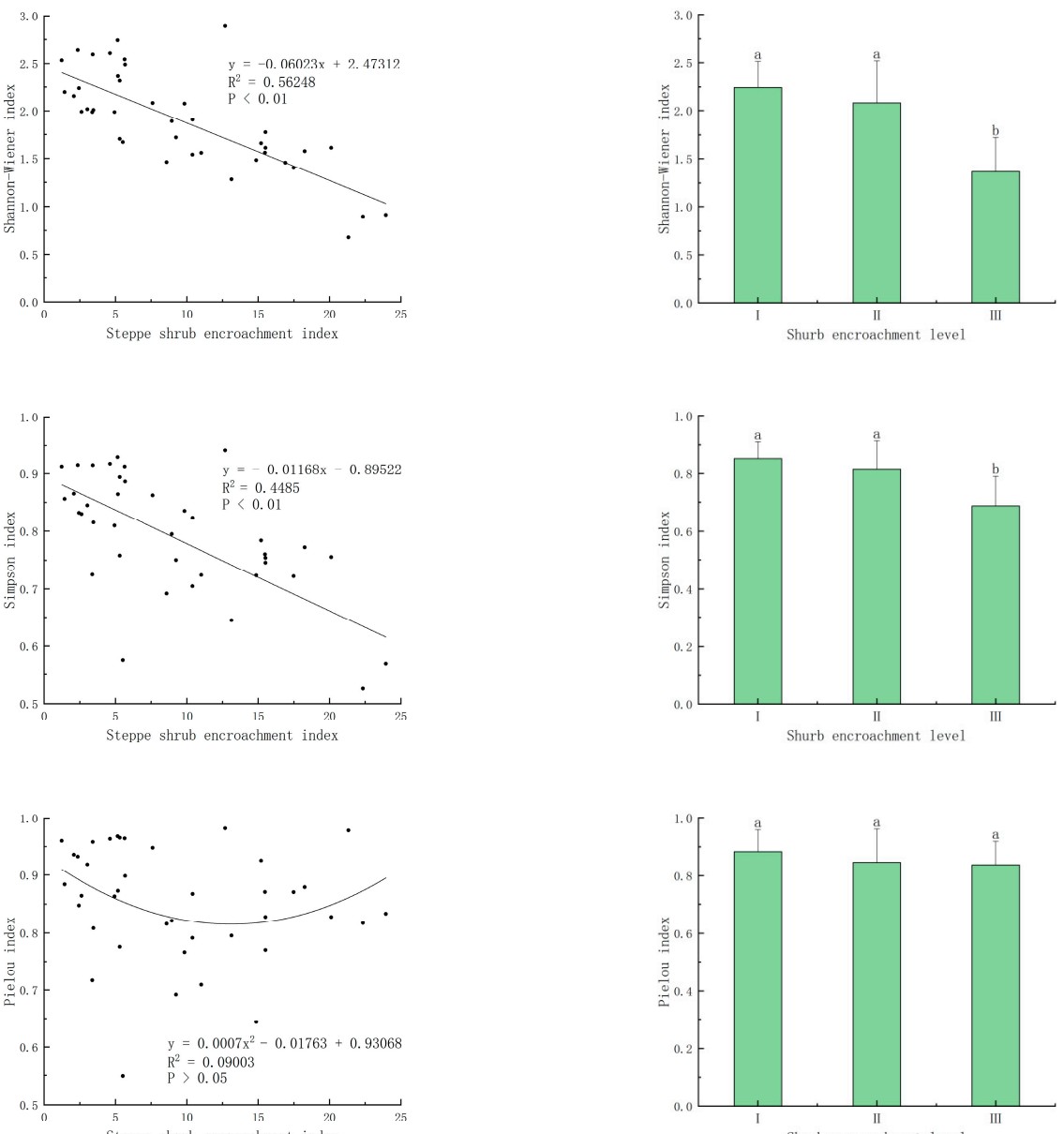

**Figure 3.** Relationship and difference in species diversity of herb communities at different shrub-level sites. Different lowercase letters indicate significant differences among sites ($p < 0.05$).

There were significant differences in the Shannon-Wiener index (H) and Simpson index (P) among the three classes of grasslands. The H-value and *p*-value of the level I and level II grasslands were significantly higher than those of level III, but there was no significant difference between them. The H-value of the level I grassland was the highest at 2.24, and that of the level III grassland was the lowest at only 1.37. The *p*-value of the level I grassland was the highest at 0.85, and the *p*-value of the level III grassland was the lowest at only 0.69. The Pielou (E) index had no significant association with the SSEI and, overall, exhibited a trend of first decreasing and then increasing. The E values of the level I grassland and level III grassland were higher overall, with a maximum value of 0.98, and the level II grassland was lower overall, with a minimum value of 0.55. There was no significant difference in the Pielou (E) index among the different levels of grassland. The E value of the level I shrub grassland was the highest (0.88), and the E value of levels II and III was the lowest (both 0.84).

### 3.2. Effects of Shrubs on Herb Community Biomass

The fitted curves representing the relationships between the SSEI and the biomass and the differences in biomass of different levels of shrub encroachment in grassland are shown in Figure 4. The results of the data analysis showed that there was no significant association between the SSEI and the aboveground biomass of shrub interspace and total aboveground biomass, but with increases in the SSEI, both of them first increased and then decreased. The aboveground biomass of shrub interspace in level II grassland was generally high, reaching its highest value of 575.3 g/m$^2$ and then gradually decreasing. The aboveground biomass of the shrub interspace of level III grassland was generally low, with the lowest value being 17.1 g/m$^2$. The total aboveground biomass of the level II grassland was higher overall, with a maximum value of 612.62 g/m$^2$; the total aboveground biomass of the level III grassland was lower overall, with a minimum value of 62.33 g/m$^2$.

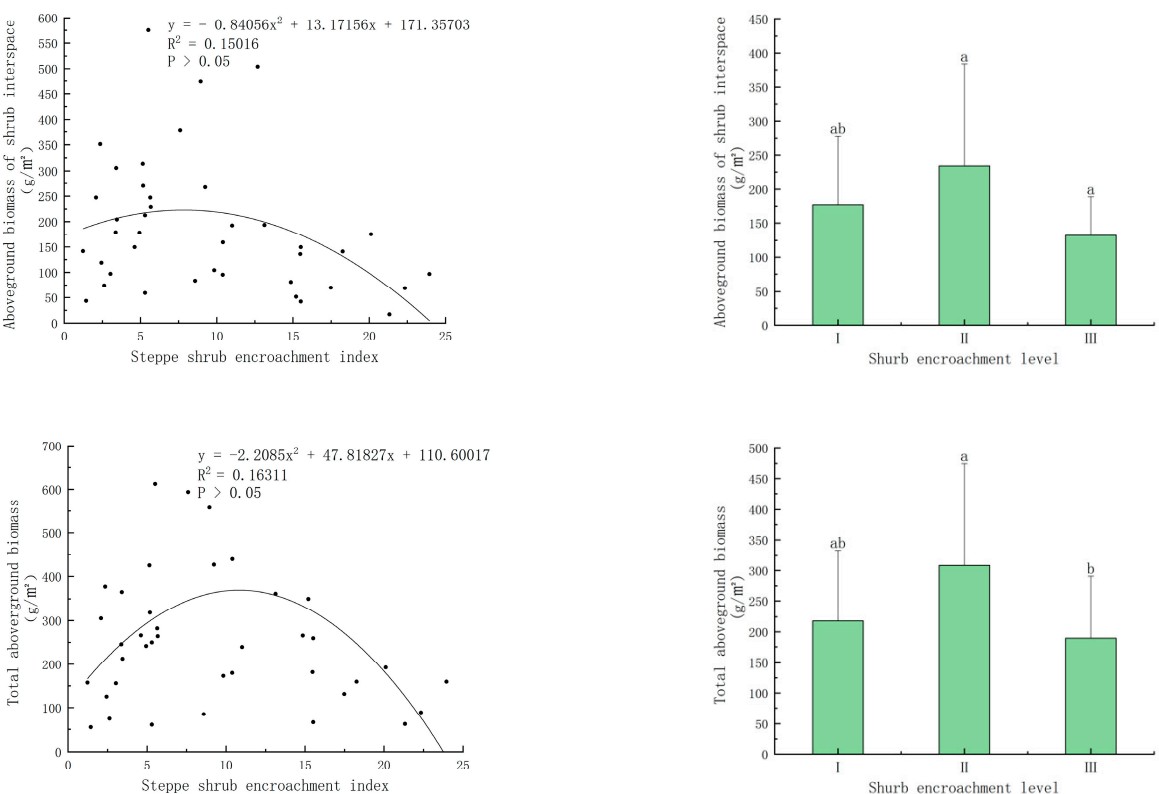

**Figure 4.** Relationship and difference in aboveground biomass at different shrub-level sites. Different lowercase letters indicate significant differences among sites (*p* < 0.05).

There were significant differences between the aboveground biomass of the shrub interspace and total aboveground biomass in different levels of grasslands. Both of them in the level III grassland were significantly lower than those in the level II shrub grassland, with the lowest values of 101.86 g/m$^2$ and 189.69 g/m$^2$, respectively.

### 3.3. Community Overall Association Analysis

The overall association between the main species in each level of grassland is shown in Table 2. The variance ratio value (VR) of the overall association of level I was greater than one, and the test statistic (W) fell within the range $\chi^2$ 0.95 (N) < W < $\chi^2$ 0.05 (N), indicating no significantly positive overall association of its main species. The variance ratio value (VR) of the overall association between levels II and III was less than one The level II test statistic (W) fell within $\chi^2$ 0.95 (N) < W < $\chi^2$ 0.05 (N), indicating no significantly negative overall association of its main species. The level III test statistic (W) fell outside $\chi^2$

0.95 (N) < W < $\chi^2$ 0.05 (N), indicating that the overall association of its main species was significantly negative.

**Table 2.** Analysis of the overall association among the main species at different shrub-level sites.

| Type of Plot | Variance Ratio | Statistic W | $\chi^2$ Threshold Value | Result |
|---|---|---|---|---|
| Level I | 1.22 | 12.23 | (3.94, 18.31) | No significant positive association |
| Level II | 0.76 | 13.68 | (9.39, 28.87) | No significant negative association |
| Level III | 0.36 | 4.32 | (5.23, 21.03) | Significant negative association |

### 3.4. Analysis of Interspecific Association

The $\chi^2$ test results of the main species in different levels of grassland are shown in Figure 5. Among 105 species pairs in level I grassland, 55 species pairs were positively associated, accounting for 52.4%; 50 species pairs were negatively associated, accounting for 47.6%; and positive species pairs were dominant. The interspecific association was not significant. Among the 105 species pairs in level II grassland, 43 species pairs were positively associated, accounting for 41.0%; 61 species pairs were negatively associated, accounting for 58.1%; and only one pair was unrelated. The negative association species pairs were dominant. The interspecific association was not significant. Among the 105 species pairs in level III grassland, 41 species pairs were positively associated, accounting for 39.0%; 60 species pairs were negatively associated, accounting for 57.1%; 4 species pairs were unrelated, accounting for 3.8%; and the negatively associated species pairs were dominant. There was one species pair with a significant positive association. The species association of different levels of grassland was relatively loose, and each species presented an independent distribution pattern.

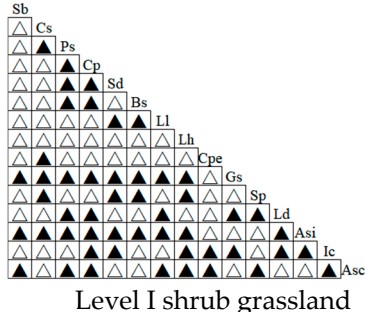
Level I shrub grassland

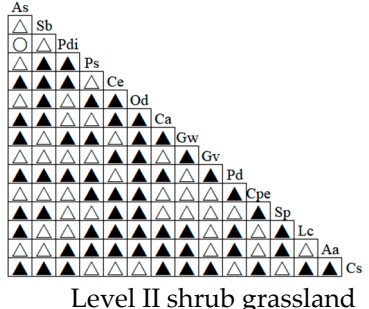
Level II shrub grassland

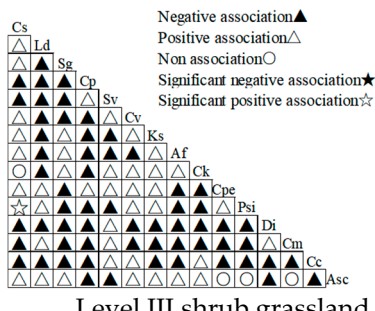
Level III shrub grassland

**Figure 5.** Half-matrix diagram of the $\chi^2$ test of different shrub-level sites. Aa: *Anemarrhena asphodeloides*; Af: *Artemisia frigida*; As: *Artemisia sacrorum*; Asi: *Achnatherum sibiricum*; Asc: *Artemisia scoparia*; Bs: *Bupleurum scorzonerifolium*; Ca: *Carex appendiculata*; Cc: *Corispermum chinganicum*; Ce: *Chamaerhodos erecta*; Ck: *Caragana korshinskii*; Cm: *Caragana microphylla*; Cp: *Cleistogenes polyphylla*; Cpe: *Carex pediformis*; Cs: *Cleistogenes squarrosa*; Cv: *Chloris virgata*; Di: *Digitaria ischaemum*; Gs: *Geranium sibiricum*; Gv: *Galium verum*; Gw: *Geranium wilfordii*; Ic: *Ixeridium chinense*; Ks: *Kummerowia striata*; Lc: *Leymus chinensis*; Ld: *Lespedeza daurica*; Lh: *Lespedeza hedysaroides*; Ll: *Leontopodium leontopodioides*; Od: *Ostryopsis davidiana*; Pd: *Plantago depressa*; Pdi: *Polygonum divaricatum*; Ps: *Potentilla supina*; Psi: *Prunus sibirica*; Sb: *Stipa baicalensis*; Sd: *Saposhnikovia divaricata*; Sg: *Stipa grandis*; Sp: *Spiraea pubescens*; Sv: *Setaria viridis*.

With the aggravation of the level of shrub encroachment, the number of positively associated species in the community decreased. The dominant species of *Cleistogenes squarrosa*—*Carex pediformis* pairs appeared in different levels of grassland, and the relationship between species changed from negative association to non-association. *Cleistogenes squarrosa*, *Carex pediformis*, and other species were mostly positively connected in level I, and when the number of negatively connected species of shrubbery increased, they were no longer associated. Comparing the association of the common species *Cleistogenes polyphylla*,

*Lespedeza daurica*, and *Artemisia scoparia* in levels I and III, the species pairs formed by *Cleistogenes polyphylla* and *Lespedeza dauricash* exhibited a more negative association, and the number of negatively associated species in level III shrub grassland was significantly higher than that in level I. This showed that the demand for the environment and resources among species in the community tended to be similar, and interspecific competition was enhanced, while the number of negatively associated species of *Artemisia scoparia* decreased with shrubbery, and the number of unrelated species increased, indicating that the species presented an independent distribution trend.

### 3.5. Spearman Rank Association Analysis

The Spearman test results of the main species in different levels of grassland are shown in Figure 6. Among the 105 species pairs in the first-class shrub grassland, 53 species pairs were positively associated, accounting for 50.5% of the total species pairs, of which 4 pairs were highly significant and 48 pairs were not significantly positively associated, accounting for 3.8% and 45.7% of the total species pairs, respectively; 51 pairs of species were negatively associated, accounting for 47.6% of the total number of species, and the association between species was not significant; there were 100 pairs of insignificant and unrelated species, accounting for 95.2% of the total number of species, and the independence of species was strong.

Among 105 species pairs in the second-grade shrub grassland, 46 species pairs were positively associated, accounting for 43.8% of the total species, and the association between species was not significant; 58 pairs of species were negatively associated, accounting for 55.2% of the total number of species, of which one pair was significantly negatively associated; there were 104 pairs of insignificant and unrelated species, accounting for 99.0% of the total number of species, and the independence between species was strong.

Among the 105 species pairs in the level III shrub grassland, 40 species pairs were positively associated, accounting for 38.1% of the total number of species, of which 3, 1, and 36 pairs were highly significant, significant, and insignificant positive associations, accounting for 2.9%, 1.0%, and 34.3% of the total number of species, respectively; 65 pairs of species were negatively associated, accounting for 61.9% of the total number of species, and the association between species was not significant. There were 101 pairs of insignificant and unrelated species, accounting for 96.2% of the total number of species, and the independence of species was strong.

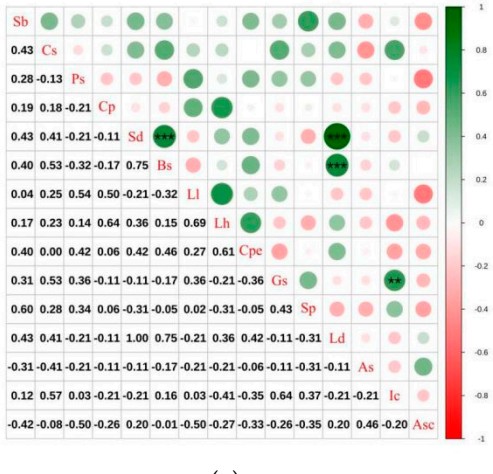

(**a**)

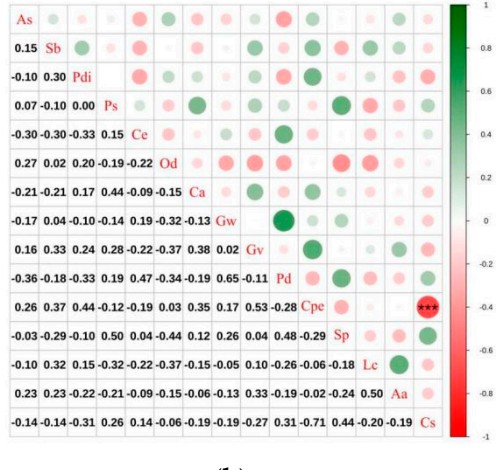

(**b**)

**Figure 6.** *Cont.*

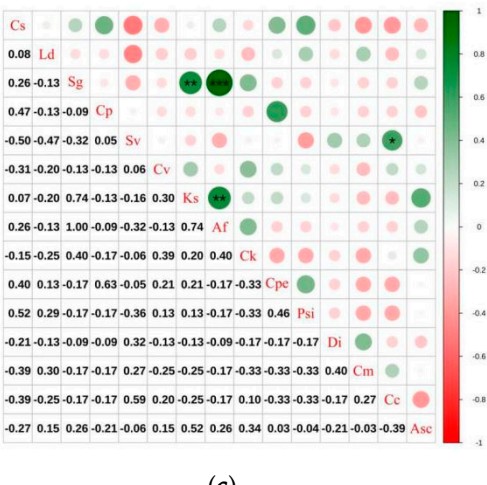

(**c**)

**Figure 6.** Semimatrix diagram of the Spearman rank association coefficient of different shrub-level sites. (**a**) Level I shrub grassland, (**b**) Level II shrub grassland, and (**c**) Level III shrub grassland. *, $p < 0.05$; **, $p < 0.01$; ***, $p < 0.001$. Aa: *Anemarrhena asphodeloides*; Af: *Artemisia frigida*; As: *Artemisia sacrorum*; Asi: *Achnatherum sibiricum*; Asc: *Artemisia scoparia*; Bs: *Bupleurum scorzonerifolium*; Ca: *Carex appendiculata*; Cc: *Corispermum chinganicum*; Ce: *Chamaerhodos erecta*; Ck: *Caragana korshinskii*; Cm: *Caragana microphylla*; Cp: *Cleistogenes polyphylla*; Cpe: *Carex pediformis*; Cs: *Cleistogenes squarrosa*; Cv: *Chloris virgata*; Di: *Digitaria ischaemum*; Gs: *Geranium sibiricum*; Gv: *Galium verum*; Gw: *Geranium wilfordii*; Ic: *Ixeridium chinense*; Ks: *Kummerowia striata*; Lc: *Leymus chinensis*; Ld: *Lespedeza daurica*; Lh: *Lespedeza hedysaroides*; Ll: *Leontopodium leontopodioides*; Od: *Ostryopsis davidiana*; Pd: *Plantago depressa*; Pdi: *Polygonum divaricatum*; Ps: *Potentilla supina*; Psi: *Prunus sibirica*; Sb: *Stipa baicalensis*; Sd: *Saposhnikovia divaricata*; Sg: *Stipa grandis*; Sp: *Spiraea pubescens*; Sv: *Setaria viridis*.

## 4. Discussion

The increase of shrub encroachment in grassland has a long history, especially in the past 160 years [38]. Shrub encroachment can affect the ecosystem from many aspects, which can lead to changes in ecosystem structure and function, reduction of net primary productivity and biodiversity, and degradation of grasslands [39]. In this study, the aboveground biomass of shrub interspace and total aboveground biomass in the moderate shrub encroachment steppe were the highest and were significantly higher than those in the heavy shrub steppe. This showed that the aboveground biomass of grassland first increased and then decreased in the process of shrub encroachment. The Shannon-Wiener index and Simpson index significantly decreased with the deepening of the shrub steppe, which was consistent with the research results of Mangani Tshepiso et al. on the semi-arid savanna in South Africa [40]. The reason for the decrease in species diversity in the shrubby steppe may be that shrub expansion occupied the living space of herbaceous plants, and the root distribution of shrubs was deeper than that of herbaceous plants, which would be beneficial for shrubs facilitating the absorption of nutrients and deep soil moisture, and the fixed value rate was higher than the death rate. Compared with herbaceous plants, they were more competitive. As a result, the rare light-loving species and the herbaceous plants with poor shade tolerance disappeared in the no-shrub grassland [41,42], and the species diversity decreased.

Shrubs can improve soil moisture and other environmental factors through crown shading to form a good microclimate, which is conducive to plant growth. The root system has a strong adsorption capacity and can concentrate organic matter in the lower soil of shrubs. Legume shrubs can also increase soil nutrients through root nodule nitrogen fixation, which has an impact on the distribution and utilization of nutrients. This phenomenon of soil nutrients gathering under shrubs is called a "fertile island" [14,43]. In the moderately shrubby steppe, the species of native herbaceous plants are abundant and affected by the "fertile island" on the community, so the level of biomass is the highest at

this time, but with the increase in shrub coverage and density, the interspecific competition between shrubs and herbaceous intensifies. Shrubs change the species composition of the community by inhibiting niche complementarity, reducing the productivity of herbs [44]. In the moderately shrubby steppe, the herb community has a low species richness, but it accounts for a large proportion of the total biomass of the community, which shows that it plays a leading role in community productivity.

The interspecific association is the association of different species in spatial distribution, which reflects the relationship between species and their adaptability to the environment. The analysis of interspecific associations can determine the interspecific relationship of the population, which is an effective way to explore community stability and the dynamic change of interspecific relationships in the process of succession. [45] A negative interspecific association usually shows the competitive effect caused by similar demands for environmental resources among species; the positive association usually shows that the species belong to different lifeforms or have different needs for environmental resources, resulting in an ecological compensation effect [46]. In this study, the $\chi^2$ and Spearman rank association coefficient tests were used to analyze the interspecific association of the main species in different levels of the shrub community in the forest grass transition area of Inner Mongolia. In the mild shrubbery environment, the community generally exhibited a positive association because, at this time, livestock feeding, trampling, and other behaviors played a major role in the growth restriction of the population [47], and resource restriction was weakened. To maintain their position and role in the community, plant populations formed a synergistic and promoting effect to resist interference in the long-term evolution process, exhibiting an affinity relationship so that the community as a whole was positively associated [42]. In moderate and severe shrubbery environments, the overall association changed to a negative association. The reason was that shrubbery reduced the survival resources of herbaceous plants, enhancing the competition between species for the same environmental resources and weakening or introducing competition between species. The vast majority of species pairs were not significantly associated, the observed association between species was loose and weak, and the distribution pattern between species was independent. This was due to the reduction in species richness, the reduction in the individual density of herbaceous plants in the community, the weakening of niche overlap between populations, the independent distribution of population individuals, and the gradual trend of non-association between populations [48], indicating that the community is in the process of dynamic succession, and the distribution among species was relatively unstructured. A close relationship had not yet been formed, and the community was in an unstable state [46]. In the future, they may continue to thicken until the grassland is completely degraded, or retrograde succession may occur, yielding a typical grassland [49].

## 5. Conclusions

Based on the field survey data, areas of the sample plot were classified based on the shrubbery level. The correlations between the shrubbery level and species diversity and population association were analyzed using statistical methods. It may provide a basis for further revealing the mechanism and impact of shrub encroachment. The final results can be summarized as follows:

(1) Shrubbery significantly affected the species diversity of the community. The SSEI was significantly negatively associated with the Shannon-Wiener and Simpson indices. The Shannon-Wiener and Simpson indices of severe shrub encroachment grassland were significantly lower than those of mild and moderate shrub encroachment grassland; that is, when the grassland developed severe shrubbery, the species diversity of the community was significantly reduced, and the species diversity of severely shrub encroachment grassland was the lowest. Mild and moderate shrubbery had no significant impact on community species diversity, while severe shrubbery significantly reduced community species diversity.

(2)　The influence of shrubbery on the aboveground biomass of grassland was not significant, the biomass showed a trend of increasing first and then decreasing, and the herbaceous population played a leading role in grassland productivity. The aboveground biomass of shrub interspace and total aboveground biomass in severely shrub encroachment grassland were significantly lower than those in mild and moderate shrub grassland.

(3)　The overall association of shrub communities in the forest grass transition area in Inner Mongolia changed from positive to negative. Most species pairs of the shrub community were not significantly associated, the interspecific association was relatively loose and weak, and the distribution pattern of each species was relatively independent. The community was in the early stage of unstable succession, and it is possible to continue to shrub encroachment or reverse succession into a typical grassland in response to the interference of human or environmental factors.

**Author Contributions:** Conceptualization, Q.S. and T.W.; methodology, Q.S.; software, Q.S.; validation, Q.S.; formal analysis, Q.S.; investigation, Q.S. and T.W.; resources, Q.S. and T.W.; data curation, Q.S.; writing—original draft preparation, Q.S.; writing—review and editing, Q.S. and T.W.; visualization, Q.S.; supervision, Q.S.; project administration, Q.S.; funding acquisition, Q.S. All authors have read and agreed to the published version of the manuscript.

**Funding:** This research was funded by [Study on Technological Innovation and High-Quality Development of Khorchin Right Wing Front Banner Grass Industry] in Inner Mongolia grant number [2021BLRD23].

**Institutional Review Board Statement:** Not applicable.

**Informed Consent Statement:** Not applicable.

**Data Availability Statement:** The data presented in this study are available on request from the corresponding author. The data are not publicly available due to privacy.

**Conflicts of Interest:** The authors declare no conflict of interest.

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
