# Peer review of "Effects of Shrub Encroachment in the Inner Mongolia Ecotones between Forest and Grassland on the Species Diversity and Interspecific Associations"

_agronomy, doi:10.3390/agronomy12102575_

Round 1

Reviewer 1 Report

The present research is interesting but the way of writing the article needs to be improved.

1. A brief description of the research should be given in the abstract. The authors present in the abstract the results of their research without describing it.

2. The research objectives are not clearly stated. Also, the hypothesis under consideration is not mentioned.

3. The authors use different terms for the same thing throught the manuscript which is very confusing for the reader.

My suggestions are indicated  in the accompanying document

Reviewer 2 Report

Line 9 – is shrub encroachment always done by native plants? In some regions, non-native plants are also a problem.

Line 15 – What does scrubbing mean?

Line 52 – improving the temp? improve for what?

Line 68 – the entire grassland ecosystem? Because like the study area, North America has different grassland regions classified by specific climate, geography and species. Specifics on where this study was done?

Line 70-71 – statement should be backed by citation.

Line 88 – is bush and shrub the same thing?

Line 88-92 – sentences contradict each other.

Line 96-97 – what is scrubbing? Increasing shrub component?

Line 104-106 – it would be nice to have the reasons for increased encroachment elaborated on. What specifically has led to increased shrubs in the region. For instance, in North America the lack of fire has resulted in increased woody plant encroachment.

Last paragraph in the introduction should have specific objectives, hypothesis or questions that will be addressed as a result of the work. These are lacking and made it challenging to assess the results.

Methods sampling section was very hard to follow.

Line 133 – check with journal format, but may need to spell out 40 to begin a sentence.

Line 133 – the use of the word “plots” is confusing here, is the plot the 10 x10 m shrub monitoring block or a larger area?

Line 138 – how was coverage estimated? Percent?

Line 139 – how does clipping the outside branches provide a biomass estimate?

Line 141 – what is sub cover?

Line 142 – what does “sampling plots were cut for each plant” mean? How was this data used to estimate biomass production?

Line 146 – why use the importance value and what is it?

Line 160 – what is scrubbing class?

Line 213 – try to avoid using a figure as the topic of your sentence. Stick to the findings and reference the figure as support.

Line 319 – what is included in total biomass?

Line 332 – “good” micro climate for what?

Line 337 – will most readers know what the “Wo Island” is?

Line 349 – what is a 2 test?

Line 344-351 – this is a good explanation that would be useful in methods. More explanation of why you used the index that were used in the analyses would be useful overall.

In general, the discussion could compare/contrast findings from additional studies, I imagine a much greater breathe of literature on shrub encroachment is available relative to what’s presented in the paper. The discussion could use more detail in genera, it is currently only three paragraphs long with the first one primarily just rehatching the results.

Conclusions – so what do your findings mean for science and society? How can they be useful going forward?

Reviewer 3 Report

The article is original and interesting from a theoretical and practical point of view, although the subject matter would be better suited for another journal (e.g. Diversity). 

1. Lines 25-27. The text is very unclear and it is not clear what meaning is given to some terms. The abstract should be clarified and a clear summary at the end with the most important result should be given.

2. Lines 132-142. This part of the methodology section is not well written. It should clearly describe each evaluation method and how the study was carried out. How many shrubs were measured for height, over what area was the number of shrubs counted, how were the indicators determined, how was cover assessed and over what area, how many shoots were used to assess biomass or were all the shrubs cut, how were the herbaceous samples taken and processed, how was the height measured, how was the density assessed? These questions need to be answered in detail, otherwise it is not clear whether the results obtained are reliable. 

3. I did not find any information in the methodology on tests of normality of the data. Which data were normally distributed and which were non-normally distributed? Judging by the Spearman correlation applied, a certain proportion of the data was non-normally distributed. 

4. The results section focuses on the correlations, but neither the results nor the discussion sections go into much detail about the types of correlations that are positive or negative, and do not answer the questions why this is so. This deficiency needs to be addressed.

5. Fig. 6 is not sufficiently informative. It does not show which parameters the correlation was calculated between. If it is the same species as Fig. 5, a reference should be provided.

6. The biggest weakness of the paper is that the information provided is completely disconnected from the ecological characteristics of the plants and is based on mathematical calculations. There is also a lack of clear conclusions on the mechanism of shrub effects on steppe plants. 

7. The idea of the paper is original and interesting, but the results are completely disconnected from the ecology and biology of the plants. I would suggest that, rather than relying solely on abstract indices, which only show trends but do not indicate their origin, we should use relevant data on plant biomass, abundance, density and other data and analyse them in relation to the traits of each species.

Reviewer 4 Report

The authors report an interesting study on the effects of shrub encroachment in Mongolian ecosystems. In particular, the authors describe shrub grassland effects in forest-grass transition areas. While the study is interesting from an ecological point of view, I found the manuscript to be very descriptive. Why didn't the authors do a temporal analysis? Why have the authors not reported the evolution of biodiversity over time? Furthermore, in my opinion, there is no link with crops. The authors do not report any association, or cause and effect with specific agronomic crops, but they give only one ecological overview. For ease, I report below the topics covered by AGRONOMY.

Subject Areas

  • Crop breeding and genetics
  • Chemistry, biology, and genetics applied to agronomy
  • Biotechnology for farming and the use of plants, plant breeding
  • Farming and cropping systems
  • Precision agriculture
  • Crop-livestock interactions
  • Crop and soil interactions
  • Soil heath and plant nutrition for sustainable agriculture
  • Agronomy of urban and peri-uban areas
  • Organic farming
  • Weed science and weed management systems
  • Industrial and bioenergy crops
  • Horticultural and floricultural crops
  • Agroecosystems and the environment
  • Sustainable development of agronomy
  • Sustainability, biodiversity and ecosystem services of bioenergy cropping systems
  • Crop physiology
  • Water management/Irrigation in agronomy
  • Agricultural meteorology (climate change)
  • Grassland and pasture improvement and agronomy
  • Food systems

Minor aspects:

Abbreviations should not be included in the abstract

Round 2

Reviewer 1 Report

No other comment

Author Response

Thank you very much for your previous comments on the revision of the article!

Reviewer 2 Report

Flow is better and the purpose of the paper is clearer. There are still errors in the writing that need to be addressed.

Author Response

Thanks for your valuable comments. We have made some changes to the article and replaced some terms.

Reviewer 3 Report

The revised article has been significantly improved, but there are still some remaining shortcomings. 

1. The source of the literature from which the statements on the impacts of climate change are taken should be cited (line 125 ff). Climate change is not causing soil drying in all regions of the world. In some regions, the opposite is true, with excess rainfall or an uneven annual distribution.

2. The methodology section has been updated with important information, but still lacks clarity and consistency. For example, it is stated that "The newborn branches of the shrubs in the quadrat were cut, dried and weighed to obtain aboveground biomass data", but it is not indicated whether the absolute dry or air-dry mass was assessed. What was considered as 'newborn branches (young shoots)? Maybe they are shoots grown during the year of the survey? 

The methodology is a crucial part of the paper and must be absolutely clear so that there is no room for ambiguous interpretation. As a reader of the paper, I first read the methodology and then I can assess whether the publication can be used as a basis for my research. If the methodology is incomplete and unclear, I cannot cite the publication because I cannot interpret the results. 

3. In my first review, I asked what method was used to find that the data were not normally distributed. Which data are non-normally distributed? Is it the area of shrubs, the height, or the calculated indices? This is very important information that must be included in the methodology, in the statistical analysis section.

4. The methodology indicates that the area of shrub cover (calculated using the ellipse formula) has been determined and that it is expressed as a percentage in Table 1. So what method was actually used to estimate shrub cover?

5. Why are the ranges of values given in Table 1 but not the mean values and standard deviation? The range is important, but it does not represent the mean values from which the whole data sample can be inferred.

6. The previous term "herb community" has for some reason been replaced by the jargon "herbaceous community". Community is abstract and cannot be either 'woody' or 'herbaceous'. We name communities according to what makes them up or what dominates them (moss, herb, shrub, etc.).

7. It was pointed out in a previous review that Figure 6 does not refer to species abbreviations. In the current version, this information is also not provided. If the species abbreviations are the same as in Figure 5, a reference should be provided so that the reader does not have to guess what the abbreviations mean.

Author Response

Thanks  for your valuable comments. Please see the attachment for specific reply.

Reviewer 4 Report

The manuscript has improved considerably. The authors followed the suggestions of the Reviewers. Now the manuscript can be published.

Author Response

Thank you very much for your previous comments on the revision of the article.